# Enteral Vancomycin to Eliminate MRSA Carriership of the Digestive Tract in Critically Ill Patients

**DOI:** 10.3390/antibiotics11020263

**Published:** 2022-02-17

**Authors:** Sophie H. Buitinck, Matty Koopmans, Rogier M. Determann, Rogier R. Jansen, Peter H. J. van der Voort

**Affiliations:** 1Department of Intensive Care, OLVG Hospital, P.O. Box 95500, 1090 HM Amsterdam, The Netherlands; s.h.buitinck@gmail.com (S.H.B.); m.koopmans@olvg.nl (M.K.); r.m.determann@olvg.nl (R.M.D.); 2Department of Medical Microbiology, OLVG Hospital, Oosterpark 9, 1091 AC Amsterdam, The Netherlands; r.r.jansen@olvg.nl; 3Department of Critical Care Medicine, University Medical Center Groningen, University of Groningen, P.O. Box 30001, 9700 RB Groningen, The Netherlands

**Keywords:** vancomycin, oral, MRSA, carriership, colonization, SDD

## Abstract

Background: Carriership with methicillin resistant *Staphylococcus aureus* (MRSA) is a risk for the development of secondary infections in critically ill patients. Previous studies suggest that enteral vancomycin is able to eliminate enteral carriership with MRSA. Data on individual effects of this treatment are lacking. Methods: Retrospective analysis of a database containing 15 year data of consecutive patients from a mixed medical-(cardio)surgical 18 bedded intensive care unit was conducted. All consecutive critically ill patients with enteral MRSA carriership detected in throat and/or rectal samples were collected. We analyzed those with follow-up cultures to determine the success rate of enteral vancomycin. Topical application of 2% vancomycin in a sticky oral paste was performed combined with a vancomycin solution of 500 mg four times daily in the nasogastric tube. This treatment was added to a regimen of selective digestive tract decontamination (SDD) to prevent ICU acquired infection. Results: Thirteen patients were included. The mean age was 65 years and the median APACHE II score was 21. MRSA was present in the throat in 8 patients and in both throat and rectum in 5 patients. In all patients MRSA was successfully eliminated from both throat and rectum, which took 2–11 days with a median duration until decontamination of 4 days. Secondary infections with MRSA did not occur. Conclusions: Topical treatment with vancomycin in a 2% sticky oral paste four times daily in the nasogastric tube was effective in all patients in the elimination of MRSA and prevented secondary MRSA infections.

## 1. Introduction

Methicillin resistant *Staphylococcus aureus* is a multidrug resistant micro-organism that is prevalent in 15–20% of hospitalized patients [1]. The prevalence in the general population in the Netherlands, which is around 1%, is low compared to other countries [1]. This is in part due to an active search policy and decontamination strategy [2]. For instance, patients who have been admitted recently in foreign hospitals are treated in isolation when they are admitted to Dutch hospitals until they prove to be without MRSA carriership. A Dutch guideline for search and elimination of MRSA carriership is available and is widely adopted in the Netherlands [3]. This guideline focusses on outpatient MRSA elimination and recommends mupirocin in the nose in combination with two oral antimicrobials, depending on the in vitro sensitivity testing. This strategy is combined with non-medical interventions such as the cleaning of clothes and the use of disinfectants.

MRSA is, similar to other *Staphylococcus aureus*, highly virulent in susceptible patients. As a consequence, critically ill patients with MRSA carriership in the digestive tract are at high risk for secondary infections like ventilator associated pneumonia and MRSA blood stream infections [4]. It is unknown whether critically ill patients identified with MRSA in their admission cultures have true colonization, which means a persistent presence of MRSA, or carriership, which implies a shorter time period with MRSA [5]. MRSA colonization can persist for months to years but can also be acquired shortly before admission at the ICU. Both, colonization and carriership, may lead to secondary infection, mostly of the respiratory tract, which necessitates preventive elimination. Infection prevention in intensive care units can be achieved by using selective decontamination of the digestive tract (SDD) [6,7]. SDD has been used primarily in European ICUs to prevent secondary endogenous pneumonia, which can occur in intensive care patients, particularly those with mechanical ventilation. SDD effectively prevent respiratory infection and bacteremia and reduces ICU mortality [8]. The fear of inducing resistant strains of microorganisms in the gut has not been proven by facts at this moment [7,9]. Half of the MRSA colonized patients carry MRSA in the digestive tract [10]. The classic SDD regime with Amphotericin B, colistin and tobramycin, is, in general, not able to eradicate MRSA from the digestive tract. To eliminate MRSA from the digestive tract, enteral vancomycin can be used [11,12,13].

In this study we describe the effects on carriership with MRSA using enteral vancomycin in individual critically ill patients.

## 2. Material and Methods

The study was performed in a teaching hospital in Amsterdam with an 18-bedded mixed medical-(cardio)surgical ICU. In a retrospective design, consecutive patients with MRSA carriership of the digestive tract were included. The patients were identified in a prospectively collected database over a period of 15 consecutive years. All patients who were treated in isolation because of proven or suspected MRSA carriership were identified. Eligible for analysis were patients with MRSA in the digestive tract, i.e., MRSA in throat or rectal culture. Excluded were patients who appeared to be without MRSA in throat or rectal cultures, MRSA carriage solely in the nose or solely in any other organ, or who had insufficient data to evaluate the primary endpoint because of a short length of stay.

The primary endpoint is the elimination of MRSA from throat and rectal cultures defined as one or more consecutive negative cultures from both throat and rectum on ICU discharge.

### 2.1. Intervention

In all patients SDD was applied to prevent secondary infections. According to the local guideline, the original SDD formulation was applied, consisting of four times daily Orabase^®^, a sticky oral paste enriched with 2% polymyxin B, amphotericin B and tobramycin [14]. For patients with MRSA, 2% vancomycin was added to the oral paste. In addition, 10 mL of a suspension containing 500 mg amphotericin B, 100 mg polymyxin B and 80 mg tobramycin was administered four times daily in the gastric tube or swallowed in patients without gastric tube [14]. In patients with MRSA 4 times daily, 500 mg vancomycin by gastric tube was added to this regime during the entire stay in the intensive care unit [13,15]. An i.v. course of cefotaxime was administered to all patients for four days but was prolonged in case of active infection with susceptible microorganisms or replaced by another antimicrobial agent in case of infection with a cefotaxime resistant microorganism. At the discretion of the attending physician, empirical antimicrobial treatment on admission was extended with ciprofloxacin i.v. or tobramycin i.v. In case of peritonitis metronidazole was added as well. Other i.v. antimicrobials can be given when previous culture results necessitate another choice. Nasal mupirocin 2% (Bactroban^®^) was applied when nasal culture appeared positive for MRSA.

### 2.2. Cultures

Cultures of the throat were performed to determine carrier state in the upper gastro-intestinal tract. For the lower tract, rectal cultures were performed. Routinely, cultures from throat and rectum were taken twice a week and tracheal aspirate three times a week for surveillance in patients treated with SDD. Culture samples taken in the context of SDD surveillance were plated on an unselective blood agar and four specific agars selecting for gram-positive bacteria, gram-negative bacteria, yeast and vancomycin resistant enterococci. In cases of (suspected) MRSA carriership, additional cultures were performed from the nose. Moreover, to improve sensitivity for MRSA, swabs from the nose, throat and rectum were sub-cultured for 18 h in a Columbia Broth supplemented with desferrioxamine, 5 mg/L and 5% sodium chloride, before plating on a selective MRSA agar.

### 2.3. Analysis

Median and interquartile range [IQR] are given for data because of the small numbers. Data were analyzed with SPSS 23.

Secondary infections with MRSA were determined by analyzing the day-to-day medical notes for diagnoses like blood stream infection, pneumonia, central line infection and other infections as structured and protocolized registration was not available.

The local medical ethical review board (ACWO OLVG) approved the study and waived informed consent due to its retrospective and observational design in accordance with Dutch and European legislation (study no. WO 18.017).

## 3. Results

Over a period of 15 years, 63 consecutive patients were identified in the database with (suspected) MRSA carriership. Fifty patients were excluded either because isolated nasal MRSA carriership was found or the ICU stay was too short to evaluate the effect of the vancomycin regimen. In 13 patients, the effects of oral and enteral decontamination with vancomycin could be evaluated. In these patients, MRSA was found solely in the throat in 8 patients and in both throat and rectum in the other 5 patients. None of these patients had MRSA at organ sites. All patients received vancomycin in the oral paste and vancomycin in the gastric tube as described in the method section.

The median age of the patients was 66 [IQR 52–78] years, their APACHE II score 21 [IQR 17.5–20.5]. Table 1 shows the medical diagnosis as the reason for ICU admission. All patients were mechanically ventilated. In all 13 patients, MRSA was successfully eliminated from the digestive tract. The time to achieve elimination of MRSA from the digestive tract was variable between 2 and 11 days with a median of 4 days [IQR 2–6.5] after ICU admission (Table 1). Secondary infections with MRSA did not occur under this antibiotic preventive strategy.

## 4. Discussion

This study shows that topical vancomycin treatment using vancomycin enriched oral paste in combination with vancomycin in the gastric tube is able to decontaminate the digestive tract of critically ill patients from MRSA in all included patients. The time to achieve decontamination was, however, variable between 2 and 11 days. In addition, this strategy prevented the development of secondary infections, such as respiratory tract infections, with MRSA.

The literature concerning the use of enteral vancomycin for the elimination of MRSA carriership in the critical care setting is scarce. Maraha et al. used oral vancomycin successfully in hospital and nursing home settings [16]. De la Cal used enteral vancomycin to reduce the MRSA prevalence in a Spanish ICU [11]. In three subsequent periods, De la Cal et al. compared no vancomycin, vancomycin in MRSA carriers and vancomycin in ‘population at risk’. The prevalence of MRSA carriership and subsequent infection decreased in these periods respectively [11]. Viviani et al. found that a 4% vancomycin containing oral gel significantly reduced the absolute carriage in comparison with a 2% gel [17]. De la Cal also used the 4% solution successfully [11]. In contrast, we have shown that a 2% concentration in the oral paste is also effective. This difference with the study from Viviani might be explained by our use of Orabase^®^, a sticky paste, instead of the gel that may have been less sticky.

Silvestri and co-workers showed that enteral vancomycin added to an SDD regime because of an MRSA outbreak reduced the rate of secondary infection caused by MRSA [12]. Their patients received 2 g of vancomycin per day by nasogastric tube but they did not add vancomycin in the oral paste [12]. Cerda et al. described the use of enteral vancomycin in addition to SDD in a burn unit over a 9-year period. They were able to reduce acquisition of MRSA with this regimen [13]. Thorburn and co-workers controlled MRSA with enteral vancomycin in 29 pediatric ICU patients [15]. They concluded that this was effective and safe.

Our study has several strengths and limitations. It is the first study that shows the results concerning the carrier state of individual patients instead of a group level. In addition, it is shown that a few days of topical treatment is enough to eliminate MRSA from the surveillance cultures. Other studies did not report on individual patients and successfulness of the decontamination strategy [11,12,13,15]. However, they evaluated the effects on the critical care population as a whole in a setting of high MRSA prevalence. The Dutch situation is one with a very low prevalence of MRSA in the population and an active search policy with subsequent isolation of MRSA carriers [3].

The size of our study is limited to 13 patients, which limits the strength of the conclusions. However, the main focus of this study is the individual dynamics of MRSA under the intervention which can better be studied in a limited number of patients. We did not collect surveillance cultures after ICU discharge which makes it unknown whether it represents suppression rather than elimination. Nevertheless, we observed a consequent effect leading to elimination of throat and rectal MRSA carriership. For a complete elimination of MRSA in a patient a bundled approach is necessary, including disinfection of skin and nasal mupirocin. Wenisch and co-workers described a successful holistic approach, including iv linezolid in combination with enteral vancomycin and topical treatment of wounds [18]. In addition to enteral vancomycin, some patients were treated with iv antibiotics as well. In all patients with iv antibiotics, cefotaxime and ciprofloxacin were used (Table 1). As MRSA is not susceptible to cefotaxime and ciprofloxacin [19], it is unlikely that these antibiotics contributed to decontamination. Some patients also received i.v. vancomycin which may have entered the gut lumen. However, the concentrations in the gut lumen that are reached by iv vancomycin are relatively low in comparison to the amount given by nasogastric tube.

The combination with SDD implies that MRSA is exposed to the combination of enteral aminoglycosides and vancomycin. Some of the MRSA bacteria are susceptible to aminoglycosides. This may have enhanced the effect of enteral vancomycin.

In addition, we did not study the effect on the susceptibility or resistance patterns of gut microorganisms, including MRSA, due to the exposure to vancomycin in the endoluminal side of the digestive tract. Enterococci and anaerobes are usually susceptible to vancomycin and oral vancomycin treatment must have had an impact on the population of these microorganisms. Previous studies did not find clinical effects, such as bacteremia or infections with vancomycin resistant enterococci, of this intervention, nor did we. Also, some resorption of vancomycin to the systemic circulation might have taken place with systemic effects. Basically, vancomycin is not absorbed from the digestive tract but critically ill patients suffer from an enhanced resorption from the digestive tract due to loosening of the tight junctions of the epithelium [20]. Previously, we described systemic vancomycin serum levels that showed a strong relation with severity of disease, measured by a SOFA score [21].

This study enables a prospective clinical trial on the effects of enteral vancomycin in patients with MRSA carriership, including its systemic effects.

In conclusion, our data suggest that adding 2% vancomycin to a sticky oral paste in combination with 4 times daily 500 mg vancomycin by nasogastric tube is able to eliminate MRSA carriership in the digestive tract, which subsequently prevents secondary infection.

## Figures and Tables

**Table 1 antibiotics-11-00263-t001:** Baseline characteristics, MRSA location and results of follow-up cultures per case.

Case Number	Age	Sex	Admission Diagnosis	APACHE II Score	Location of MRSA	Time until Negative Cultures (Days)	No. of Follow Up Cultures	No of Negative Follow Up Cultures	IV Antibiotics
Case1	57	M	Pneumonia	18	Throat	4	9	9	Cefotaxime; ciprofloxacin; vancomycin
Case2	88	F	Abdominal sepsis	23	Throat, rectum	11	6	2	Cefotaxime; ciprofloxacin
Case3	70	F	Abdominal sepsis	19	Throat	4	6	6	Cefotaxime; ciprofloxacin; vancomycin
Case4	80	M	Abdominal sepsis	24	Throat	6	3	2	Cefotaxime; ciprofloxacin
Case5	65	M	Cardiogenic shock	32	Throat	2	2	2	Cefotaxime; vancomycin
Case6	76	F	Respiratory failure	27	Throat, rectum	7	8	6	Cefotaxime; vancomycin
Case7	54	M	ARDS	21	Throat	4	3	3	Cefotaxime; vancomycin
Case8	50	M	Pneumonia	14	Throat, Rectum	7	4	4	Cefotaxime; ciprofloxacin; vancomycin
Case9	66	M	Respiratory failure	24	Throat, Rectum	2	1	1	Cefotaxime
Case10	41	F	Abdominal sepsis	9	Throat	2	3	3	Cefotaxime; ciprofloxacin; vancomycin
Case11	44	M	Pneumonia	18	Throat	2	4	4	Cefotaxime; ciprofloxacin; vancomycin
Case12	85	M	Abdominal sepsis	27	Throat, Rectum	3	3	3	Cefotaxime; ciprofloxacin
Case13	69	M	Respiratory failure	17	Throat	2	2	2	Cefotaxime
Median (IQR)	66 (52–78)			21 (17.5–20.5)		4 (2–6.5)	3 (2.5–6)	3 (2–6)	

## Data Availability

Data are available on request.

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
