# Peer review of "Enteral Vancomycin to Eliminate MRSA Carriership of the Digestive Tract in Critically Ill Patients"

_antibiotics, 2022, doi:10.3390/antibiotics11020263_

Round 1
Reviewer 1 Report
Dear Authors
The article "Enteral vancomycin to eliminate MRSA carriership of the digestive tract critically ill patients" concerns an important issue of modern medicine.
The work follows the layout typical of scientific work, but it is very sparing in words, so it does not accurately describe the topic under discussion.
Please refer to the following comments:
Individual sections require expansion. In present versions, they are limited and do not, in a sufficient way, explain readers main topic.
There is no information about the dose that patients take. There is only information about concentration 2% paste, but no information about vancomycin dose.
Was MRSA removal due to vancomycin paste or as described in the Intervention section a result of several antibacterial drugs co-therapy?
The main objection, which is also noticed by the author, is the very small patient group included in this study, which makes it impossible to translate the results into clinical practice. The authors did not prove that this is due only to the paste in vancomycin.
This work should be described as communication rather than the article.
I have to say that the work requires a major overhaul before publication and is unsuitable for publication in the Antibiotics journal as it stands.
Author Response
We thank the reviewer for the valuable comments. Here we provide a point by point response in italics.
Reviewer 1: The article "Enteral vancomycin to eliminate MRSA carriership of the digestive tract critically ill patients" concerns an important issue of modern medicine.
The work follows the layout typical of scientific work, but it is very sparing in words, so it does not accurately describe the topic under discussion.
Please refer to the following comments:
Individual sections require expansion. In present versions, they are limited and do not, in a sufficient way, explain readers main topic.
Answer: we have expanded the original text in most sections to describe the topics more in depth. The main body has been increased with around 400 words.
There is no information about the dose that patients take. There is only information about concentration 2% paste, but no information about vancomycin dose.
Answer: the method section mentions the dose, which is 4 times daily 500 mg. The reviewer may have overlooked this information.
Was MRSA removal due to vancomycin paste or as described in the Intervention section a result of several antibacterial drugs co-therapy?
Answer: we added information concerning antibacterial co-therapy which was cefotaxime and/or ciprofloxacin. As MRSA is not susceptible for cefotaxime and usually not for ciprofloxacin, it is unlikely that this therapy has had impact on MRSA carriership. We explained this in the text. Also, in some patients vancomycin i.v. was used but, in relation to the oral vancomycin, this will not have added significantly to the elimination of MRSA.
The main objection, which is also noticed by the author, is the very small patient group included in this study, which makes it impossible to translate the results into clinical practice. The authors did not prove that this is due only to the paste in vancomycin.
Answer: The main focus of this paper is to study individual dynamics of critically ill patients with MRSA, which gives additional information to the population studies that have been published. The 13 patients is a rather small sample but it also gives the possibility to study the individual dynamics. The successful decontamination in all patients is a strong signal, in combination with previous published articles, that enteral vancomycin can eliminate MRSA carriership. We have expanded the discussion on this point.
This work should be described as communication rather than the article.
Answer: we think that this should be a decision made by the editor.
I have to say that the work requires a major overhaul before publication and is unsuitable for publication in the Antibiotics journal as it stands.
Answer: we have made significant revisions to apply to the comments of the reviewer.
Reviewer 2 Report
The authors of the study demonstrated that enteral vancomycin as 2% paste and 500mg orally 4 times daily was successful at eradicating MRSA from the digestive tract of patients. The novelty here is that it was at a patient-specific level, rather than looking at the hospital ward incidence and endemicity.
The manuscript is well written, though short. Despite it being short, it provides additional information on the effectiveness of oral vancomycin in eradicating MRSA from the digestive tract.
Some specific comments:
1) Maybe its the formatting of the draft, but ensure that the genus and species of S. aureus when listed are italicized throughout the manuscript.
2) Did the patient receive any anti-MRSA IV agents? It is noted what the standard antibiotic regimens are but also states changes can be made by the prescriber. It would be interesting to know if any of these 13 patients received IV treatment with an antibiotic with MRSA activity. This could be presented in a column on table 1 with the name of the agent if one was received. While it is likely that IV antibiotic concentrations achieved in the colon would probably vary depending on the agent, it would be good to provide this information if available.
3) Were patients treated with oral vancomycin during the entire ICU stay? Duration of decolonization antibiotics wasnt clearly stated, this could be added.
4) For patients where the number of follow-up cultures were greater than number of negative cultures, e.g., patient case2, where 6 cultures were taken, and 2 were negative and that it taking 11 days until negative. In the manuscript, it can be mentioned that some patients did not become negative as quickly as others, highlighting some patients may require longer treatments.
5) May want to include in the discussion section the theoretical effect oral vanco may have on Enterococci and VRE development. I believe some studies that were cited in the manuscript, where oral vanco was used did not find increased VRE, this could be mentioned.
Author Response
We thank the reviewer for the valuable comments. We have a point by point response written in italics.
Reviewer 2: The authors of the study demonstrated that enteral vancomycin as 2% paste and 500mg orally 4 times daily was successful at eradicating MRSA from the digestive tract of patients. The novelty here is that it was at a patient-specific level, rather than looking at the hospital ward incidence and endemicity.
The manuscript is well written, though short. Despite it being short, it provides additional information on the effectiveness of oral vancomycin in eradicating MRSA from the digestive tract.
Some specific comments:
1) Maybe its the formatting of the draft, but ensure that the genus and species of S. aureus when listed are italicized throughout the manuscript.
Answer: we have now used italics for Staphylococcus aureus as requested by the reviewer.
2) Did the patient receive any anti-MRSA IV agents? It is noted what the standard antibiotic regimens are but also states changes can be made by the prescriber. It would be interesting to know if any of these 13 patients received IV treatment with an antibiotic with MRSA activity. This could be presented in a column on table 1 with the name of the agent if one was received. While it is likely that IV antibiotic concentrations achieved in the colon would probably vary depending on the agent, it would be good to provide this information if available.
Answer: we have now added this information in the results section in table 1. Most patients received cefotaxime with or without ciprofloxacin. However, these antimicrobial agents are not effective against MRSA. We have added this in the discussion section as well. Some patients also received i.v. vancomycin which may have entered the gut lumen. However, the concentration due to iv vancomycin in the lumen is relatively low in comparison to the much higher dose given by nasogastric tube.
3) Were patients treated with oral vancomycin during the entire ICU stay? Duration of decolonization antibiotics wasnt clearly stated, this could be added.
Answer: oral vancomycin was given during the entire ICU stay. We have added this information in the methods section.
4) For patients where the number of follow-up cultures were greater than number of negative cultures, e.g., patient case2, where 6 cultures were taken, and 2 were negative and that it taking 11 days until negative. In the manuscript, it can be mentioned that some patients did not become negative as quickly as others, highlighting some patients may require longer treatments.
Answer: we have added this information in the results and discussion section.
5) May want to include in the discussion section the theoretical effect oral vanco may have on Enterococci and VRE development. I believe some studies that were cited in the manuscript, where oral vanco was used did not find increased VRE, this could be mentioned.
Answer: we have added this point to the discussion.
Reviewer 3 Report
Overall, the authors use the term carriership rather than colonization in some of the paper, would clarify why and stay consistent as this can be confusing for some readers. Refer here: https://pubmed.ncbi.nlm.nih.gov/25595843/
Abstract:
Background:
- Interesting point about the Dutch guideline any other institutions/countries that adapt the same policy. However, would highlight how this paper can be important to other countries with very different policies to increase the external validity of the study
- The authors should discuss at some point the risks associated with SDD: https://www.ncbi.nlm.nih.gov/pmc/articles/PMC3484923/
- Overall, the background is too brief and does not provide sufficient background for the reader. If this is submitted as a case report as I understand then this is appropriate.
Method:
- There is a double exclusion: MRSA in throat or rectal cultures, MRSA carriage solely in the nose (it contradicts the inclusion)
- There is no mention of method of MRSA detection used, need to be clear about how and what entitled a positive MRSA in GI: https://www.sciencedirect.com/science/article/pii/S1198743X14606315
- What about patients who had MRSA positivity in GI and other sites
- Explain other confounding factors to GI: immune related factors, pregnancy, age, foreign born etc
- Excellent description of intervention but 0 citation, need to support where these doses come from even if it’s a standard protocol (then say that it is and try to cite it)
Results:
Can expand on table with confounders to MRSA recolonization
Also add risks associated with SDD and if this occurred in any patient and acknowledge that some risks exist in the literature, why is this not standard practice from an infection control prospective
Small number of patients so median and IQR are more appropriate
Conclusion: due to the small number I would be more conservative and say that this study “suggest” or “SDD may be”
Author Response
We thank the reviewer for the valuable response. We have written our answers in italics.
Reviewer 3: Overall, the authors use the term carriership rather than colonization in some of the paper, would clarify why and stay consistent as this can be confusing for some readers. Refer here: https://pubmed.ncbi.nlm.nih.gov/25595843/
Answer: we understand the difference between carriership and colonization. We have discussed this issue in the introduction and we have now consequently used carriership in the manuscript.
Interesting point about the Dutch guideline any other institutions/countries that adapt the same policy. However, would highlight how this paper can be important to other countries with very different policies to increase the external validity of the study
Answer: we have now added a more detailed description of the recommendations from this guideline in the introduction. The link to the guideline is updated and now links to the English version of this guideline.
The authors should discuss at some point the risks associated with SDD: https://www.ncbi.nlm.nih.gov/pmc/articles/PMC3484923/
Overall, the background is too brief and does not provide sufficient background for the reader. If this is submitted as a case report as I understand then this is appropriate.
Answer: we have discussed in the introduction more in detail now the background of SDD and the risks associated with its use.
Method:
There is a double exclusion: MRSA in throat or rectal cultures, MRSA carriage solely in the nose (it contradicts the inclusion)
There is no mention of method of MRSA detection used, need to be clear about how and what entitled a positive MRSA in GI: https://www.sciencedirect.com/science/article/pii/S1198743X14606315
Answer: The methods section mentions that the routine surveillance cultures taken on admission in all patients in the ICU are also screened for (possible) MRSA. When a suspicion rose, additional cultures were collected as described with Columbia Broth and supplements. We think that this cannot be explained better than as it is.
What about patients who had MRSA positivity in GI and other sites
Answer: none of these patients had organ sites positive for MRSA. We have included this information in the results section now.
Explain other confounding factors to GI: immune related factors, pregnancy, age, foreign born etc
Answer: the main risk factor for MRSA colonization in the Netherlands is working with or living nearby livestock, especially pigs. Other risk factors as mentioned by the reviewer is unlikely in the Netherlands. We do not have information about livestock or other risk factors in the included patients and, unfortunately, we cannot add this information to the manuscript.
Excellent description of intervention but 0 citation, need to support where these doses come from even if it’s a standard protocol (then say that it is and try to cite it)
Answer: we have now added two references concerning SDD and also concerning the vancomycin intervention.
Results:
Can expand on table with confounders to MRSA recolonization
Answer: we don’t exactly understand what the reviewer is aiming at with ‘confounders to MRSA recolonization’. Anyway, we do not have detailed clinical information to add to the manuscript.
Also add risks associated with SDD and if this occurred in any patient and acknowledge that some risks exist in the literature, why is this not standard practice from an infection control prospective
Answer: in the introduction we explained the use, the benefits and the fear for resistance of SDD.
Small number of patients so median and IQR are more appropriate
Answer: we replaced the mean and SD for median and IQR.
Conclusion: due to the small number I would be more conservative and say that this study “suggest” or “SDD may be”
Answer: we have replaced ‘shown’ into ‘suggest’ in the conclusion.
Round 2
Reviewer 1 Report
Dear Authors
Thank you for your response.
Reviewer 3 Report
Edits are appropriate, what I mean by confounders are potential factors that have the likelihood to change the outcome. Hence, they should be addressed